# Large-Scale Tactile Detection System Based on Supervised Learning for Service Robots Human Interaction

**DOI:** 10.3390/s23020825

**Published:** 2023-01-11

**Authors:** Fábio Cunha, Tiago Ribeiro, Gil Lopes, A. Fernando Ribeiro

**Affiliations:** 1Industrial Electronics Department, University of Minho, 4800-058 Guimarães, Portugal; 2Centro ALGORITMI, University of Minho, Campus Azurém, 4800-058 Guimarães, Portugal; 3INESCTEC, University of Maia—ISMAI, 4475-690 Maia, Portugal

**Keywords:** robotics, service robots, human-machine interface, touch sensor, Machine Learning, Artificial Neural Networks

## Abstract

In this work, a large-scale tactile detection system is proposed, whose development is based on a soft structure using Machine Learning and Computer Vision algorithms to map the surface of a forearm sleeve. The current application has a cylindrical design, whose dimensions intend to be like a human forearm or bicep. The model was developed assuming that deformations occur only at one section at a time. The goal for this system is to be coupled with the CHARMIE robot, a collaborative robot for domestic and medical environments. This system allows the contact detection of the entire forearm surface enabling interaction between a Human Being and a robot. A matrix with sections can be configured to present certain functionalities, allowing CHARMIE to detect contact in a particular section, and thus perform a specific behaviour. After building the dataset, an Artificial Neural Network (ANN) was created. This network was called Section Detection Network (SDN), and through Supervised Learning, a model was created to predict the contact location. Furthermore, Stratified K-Fold Cross Validation (SKFCV) was used to divide the dataset. All these steps resulted in Neural Network with a test data accuracy higher than 80%. Regarding the real-time evaluation, a graphical interface was structured to demonstrate the predicted class and the corresponding probability. This research concluded that the method described has enormous potential to be used as a tool for service robots allowing enhanced human-robot interaction.

## 1. Introduction

In the following years, collaborative robots such as CHARMIE [1] are expected to be a fundamental part of geriatric care, domestic and medical assistance. CHARMIE is a healthcare and domestic service and assistive robot capable of performing tasks in non-standardised environmental settings. Some examples of how this robot can help in domestic and healthcare environments is doing some errands, picking objects from the floor, transport difficult objects (heavy or dangerous) and be a company robot to stimulate the mental component of the elderly. The goal is to help taking care of human’s daily lives, for tasks such as helping doctors in patients’ recovery [2]. Safety and correct interaction will be essential when designing these types of robots because, no matter how precise the robot’s movement may be, collisions avoidance cannot be guaranteed. Therefore, the implementation of touch sensors is crucial for the robot to safely interact with various agents and the environment. The large-scale tactile detection system will help with two main components. The first, as physical contact is unavoidable, the robot’s structure must decrease the inherent danger and therefore not be rigid.

The second, this system can also be used as an instruction keyboard where users can interact with the robot and interpret commands according to the place where it was touched. This method provides a solution for users that cannot communicate with a robot through speech, such as mutism, to interact and be able to make the most out of collaborative service robots.

### Previous Research

The detection and localization of touch and the different types of forces that a body is exposed to are essential to enable safe human-robot interaction. In order to reproduce the physical interaction in robots, there are different studies describing the development of tactile detection systems, which may have a wide range of purposes.

In recent decades, touch sensors have been primarily focused on robotic fingers and hands [3], however, it has been extended to the entire body. A variety of technologies have been used to improve the touch detection capability of robots, with electrical resistance [4], capacitance [5], electromagnetic induction [6], temperature [7], and others. These designs are highly accurate and are considered adequate for detecting physical contact. All these devices had difficulties in their large-scale production because they needed to add multiple sensors and electronic data acquisition. Since the size is a significant variable, integrating these components becomes a challenging process. Moreover, most advanced applications integrate several types of sensors, as in the case of the HEX-O-SKIN [8]. This structure consists of a small size printed circuit board shaped like a hexagon, equipped with multiple discrete proximity, acceleration, and temperature sensors. This HEX-O-SKIN model presents a control architecture to merge the data obtained from the different modules, making it possible to reproduce certain human sensations. It is also a prototype where replicating and acquiring data is very costly.

Another method presented in [9], is the design and hardware fabrication of an arm and hand for human-robot interaction. This robotic arm features two inflatable force sensor modules that absorb impact and provide feedback. Although this is an effective method for touch detection, it has a limitation. This model has not been able to find the cell of contact because the sensors used, only provide the magnitude of the force.

Using two cameras (OV2710 2MP USB CAMERA LOW-LIGHT SENSITIVITY) at the two ends of an elongated structure and a transparent bone, the authors [10,11] developed a tracking three-dimensional movement of internal markers under interaction with the surrounding environment. This tactile system called Muscularis is vision based. It has a set of white points internally called markers, uniformly distributed. This model is useful for analysing the detailed displacements of all the markers and estimating the contact area when a contact is made. However, the disadvantage of this approach is the use of two cameras in the robotic arm region. Therefore, to be implemented in all parts of the robot, the number of cameras is considerably high. This integration would increase the cost and the prototype’s mechanical complexity and turning it impracticable. It would occupy each internal part of the robot since the cameras would need to observe all the markers. This latter limitation makes it impossible to couple this setup on CHARMIE [1].

Despite being a system with many methods developed by the scientific community, the implemented structures have the limitation of addressing a specific problem, which results in a valid method for a specific region of the robot. Yet, the currently proposed method can be applied to any area of a robot, including legs, torso, head and other parts. Furthermore, the learning process does not need the characteristic equations of the sensors, thus reducing its complexity.

## 2. Materials and Methods

The developed sensor has a shape like the human arm (sleeve), and it is made of silicon to be flexible. Although Muscularis has the limitations present earlier, a replica (vision sleeve) shown in Figure 1, was built to allow a comparison with the proposed method (test sleeve). The interface demonstrated in Figure 1 shows the image captured by the two cameras on both sides of the sleeve, top and bottom and where the algorithm detects a touch event. In addition, on the bottom right it is shown the outside perspective of where the sleeve is being touched. These sleeves are divided in 64 sections and the cameras on the vision sleeve were strategically positioned to make it possible to observe the whole internal part of the test sleeve.

### 2.1. Test Sleeve Design

The sleeve was designed to have a cylindrical physical constitution that resembles the Human forearm. Its main dimensions are shown in Figure 2.

Figure 2 shows four measurements, where M1 describes the external diameter of the sleeve centre, M2 refers to the external diameter of the ends, M3 defines the total length and M4 indicates the distance of the sleeve that is used to fix the structure to the jig. The latter feature is a sleeve piece that does not detect any contact. The numerical values of these dimensions are shown in Table 1.

The test sleeve design required a set of PLA parts printed by a 3D printer, divided into four categories: internal, external, coalition and support. These parts were used to create the mould to furthermore create the silicone sleeve. The internal and external parts had similar shapes but different dimensions, so they were 10 millimetres apart, consisting of the sleeve thickness. The outer structure uses a set of embedded lines to define the section boundaries. The connection jigs had the purpose of interconnecting the internal and external parts, where they present distinct shapes since the upper form needed a set of holes that allowed the silicon to be applied. Finally, the end part was used to ensure that endpoints were attached to the main structure.

After assembling the structure, the uncoloured RTV-2 HB Flex 0020 silicon was poured inside. Before the silicon was cured, piezoelectric sensors were added at the end of each four sections. The piezoelectric sensors are connected to the ADC inputs of microcontroller, an Arduino Mega. Figure 3 represents the data acquisition system hardware.

In addition, in parallel with each sensor, there is a 1 MΩ resistor in order to limit the current and voltage produced by the piezo sensor to preserve the analog port. As the model exhibited 64 areas, 16 sensors were used. This prototype is shown in Figure 4.

From the analysis of the sensors’ behaviour, it was noticed that Section 5 was the one with the worst detection. Thus, it was concluded that the application of a sensor at the end of 4 sections is ideal.

The sleeve presents a non-linear cylindrical structure and there is a variation of the perimeter along the sleeve (see Figure 5). This particularity caused a variation in the distance of the sensors in the horizontal part of the sensors array, where the ends of the sleeve the sensors are horizontally distant around 5.3 cm. In the central area, where the sleeve has the greatest perimeter, the sensors are 8.3 cm apart. Vertically, the sensors have a similar distance in all areas of the array, with an average interval of 8 cm.

### 2.2. Setup for the Artificial Neural Network (ANN)

The work presented in [11], explored images based on Deep Learning techniques. However, this research uses a set of numerical values output by the sensors to classify the sleeve interaction with the environment. Since the sensor information is continuous sequential data, the ANNs were the most appropriate structure to recognize the contact zone. To build this model it was necessary to group a set of data to train and validate the Neural Network, which was based on three steps: extracting the relevant data, pre-processing, and clustering the data. Details are listed below.

(1) Data Extraction: The developed prototype was designed to have a set of sensors embedded in a malleable material. It was necessary to constantly acquire the numerical values presented on each sensor, allowing the model to perceive the sleeve’s interaction and evaluate it in real-time. As a result, it became necessary to add a data acquisition process and therefore it was implemented a microcontroller with ADC inputs. Since the piezoelectric sensors were highly sensitive and unstable, it was essential to apply a set of processes that would stabilize the values. It was then concluded that extracting the maximum sensor value in a certain period of time provided better results. However, as a real-time classification was the objective, this period could not be too long since it would cause a delay in the model prediction when the sleeve was deforming. The best values were achieved within a period of 200 milliseconds.

(2) Pre-processing: The piezoelectric sensor signal was inserted into an ADC input with 10-bit resolution and the digital output value was between 0 and 5 V. After reading the sensor values, an average of three consecutive readings was used to produce a stable and more accurate reading. A threshold was also implemented to avoid involuntary triggering on low-value readings. The data are normalized to a value between 0 and 1 at the start of the programme’s execution, in order to facilitate the model training process. In this conversion, the highest numerical value is obtained from the readings of the 16 sensors and the remaining values are divided by this value.

(3) Data clustering: The model used a Supervised Learning method to teach the Neural Network, which involved structuring a dataset with the appropriate classifications. During the process of applying the deformations on the test sleeve on the outside (touching it with the fingertip for example), the data from the sensors were collected, as well as the respective section where the contact was made. In this work, 20 touch trials were applied through all 64 sections, totalling 1280 readings. Each reading had 16 values that matched the number of sensors located on the surface. In addition, 20 readings were also collected when there was no touch at all, in a way that the model also indicates this particular situation. This process resulted in a dataset that in total represented by 1300 readings.

### 2.3. Section Detection Network (SDN) Architecture

This prototype was designed to detect the deformation location, where the type of contact was presumed to be unique. The model then only needs to predict a single section when facing a deformation. A fully connected feed forward neural network was used. The input vector has one dimension with 16 elements representing each sensor. Table 2 provides details for each network layer, where four dense layers are defined. The first three layers have the ReLU activation function and the last one, which represents the final layer, contains the Softmax function. Since this was a classification problem, the final layer presents 65 nodes, where 64 correspond to the number of sections outlined in the sleeve and the last one characterizes the lack of touch.

## 3. Results

This section describes the process of real-time training, validation and evaluation of the model and aims to show the potential of this application.

### 3.1. Stratified K-Fold Cross Validation

As the training dataset for the developed model was not large enough, it became necessary to adopt a more efficient method considering the collected data. The approach used was the Stratified K-Fold Cross Validation (SKFCV). This procedure consists of homogeneously partitioning the whole dataset into k different subsets, with the main purpose of getting several subgroups that represent all the classes. The average distribution of each class in the different groups is approximately equal [12]. The *k-1* subsets are used to train the network, while the rest is used to validate the training. This process is performed *k* times sequentially, changing the validation subset.

Before implementing this methodology, it was necessary to split the dataset into two sub-datasets: the training data and the test data. 20% of the dataset was used to form the test data (260 readings) only to evaluate the performance of the predictive model after learning. The remaining 80%, which corresponds to 1040 readings, were taken for the training and validation procedure. This 80% performance is considered good based on all prototypes and tests carried out. This is because three models with several methods were developed to make this structure feasible.

In this experiment, *k* was set to five, based on the research presented in [11], meaning that the training data subset was split into five subsets. Each subset displayed 208 readings. This process is demonstrated in Figure 6, which shows the workflow.

### 3.2. Training and Validation of Neural Network

During the training phase, the SKFCV acted on the training data, dividing them into five subsets, where one is used to validate the network and the others for training. The validation subset over the iterations, was changed in position on the dataset, so that it was possible to use all the data to validate the network.

For the definition of hyperparameters such as learning rate, epochs and batch size, a trial-and-error process was used, where the values applied were the most common for this type of application. Table 3 presents the final values applied to the model.

After the model development, system analysis in the training and testing phase was carried out. These steps used the previously acquired dataset, which was divided into subsets, in a way similar to the other methods described. Figure 7a,b present, respectively, the accuracy and loss of the training and validation data, regarding the number of epochs. Five graphs corresponding to this process were extracted from the SKFCV. Figure 7 shows only the average of all iterations.

Based on these figures, it can be concluded that the Neural Network works well for the training data. This implementation also considered the training time, which was around 2 min.

### 3.3. Training and Validation of Neural Network

This section evaluates the performance of the trained and validated models in KFCV with the introduction of Stratified Sampling. The initial datasets were divided into two subsets: training and test data. The training data was used to teach the model to understand the data placed in its inputs, so that the model learns to process the exposed information. The test data was used to evaluate the model performance for data that is unknown by the model. The most common measure of a performance evaluation using the test data was accuracy.

From the former model as well as the definition of all hyperparameters, an accuracy of slightly over 80% was achieved, which is considered a good performance. This setup tested touches from several different people with different intensities and the system managed to keep the same accuracy.

### 3.4. Real-Time Evaluation

This evaluation consisted of applying the data provided in real-time to the Neural Network to see how well the model classification performed. This analysis is supported by the visualization of a graphical interface created to simplify section identification. This interface consists of a table where the 64 sections are represented. The predicted class only changes colour in the table when the probability is higher than 75%. It also prints the achieved probability as well as the predicted class. The degree of shade in the area that represents the predicted class varies according to the resulting probability, with the greatest shade characterizing the maximum numerical value. This step is documented in the video in Appendix A. Figure 8a,b show examples of model prediction for a given contact, where it is possible to notice the difference in the shade of the predicted class based on the probability, as well as the predicted class and the probability.

After analysis of this model, it was evaluated with an average performance since it represents several times the class classification in the neighbouring region. Therefore, it was concluded that this method has potential, as it detects small deformations and does not require significant force.

## 4. Conclusions

The purpose of this project was to build a safe interaction interface between Humans and collaborative service robots, more specifically, a real-time touch detection system to enable CHARMIE to identify possible collisions when its forearm had been touched. This could translate to the user selecting a specific area so that the robot would perform a specific task or even for safety purposes, creating a safe human-robot interaction. A feature that can be implemented is the definition of a set of tasks in the matrix present in the test sleeve that the robot intends to do to assist people with speech difficulties. This system was intended to correlate low cost and effectiveness, as there are similar solutions in the scientific community, but using a large array of sensor combinations. Thanks to Machine Learning, a model was created to predict the contact area. Through the results generated in each of the steps, it was possible to observe the model’s good performance.

This model has twice the sections compared to the previous research described, making the entire test sleeve surface detectable. This system proved to have a short training time, good performance and easy to couple this structure to any type of robot with a similar manipulator or end-effector shape. It can also be used on any type of surface, flat or round.

This implementation resulted in a predictive model with an accuracy of 80%, which in real time showed a medium performance, representing by several times the neighbouring sections. Despite not being a desirable result, this model has a huge potential. This is because only 16 sensors were used to cover the entire surface of the sleeve, the extraction of the sensors values to constitute the dataset was not appropriate and only 20 readings per class were used. Even though there are all these limitations, it was possible to obtain a model with reasonable results.

As for the final model characterization, new processes need to be introduced to improve the predictive capacity of the model. The improvements planned for future work are to increase the dataset, considering the application of contact in several areas simultaneously. In addition, a regression model will be implemented, which makes it possible to know the deformation depth.

## Figures and Tables

**Figure 1 sensors-23-00825-f001:**
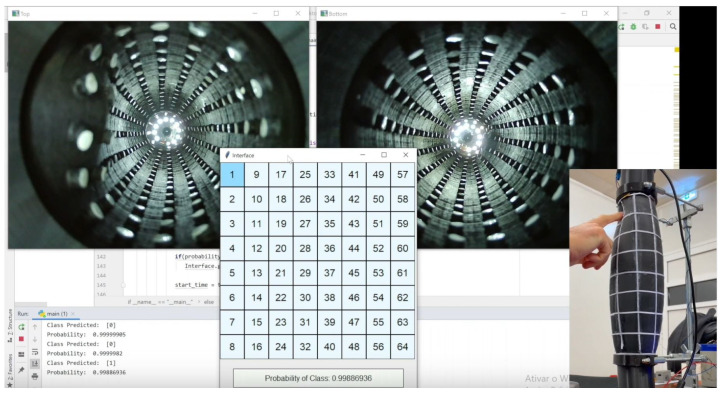
Real-Time classification of the vision sleeve.

**Figure 2 sensors-23-00825-f002:**
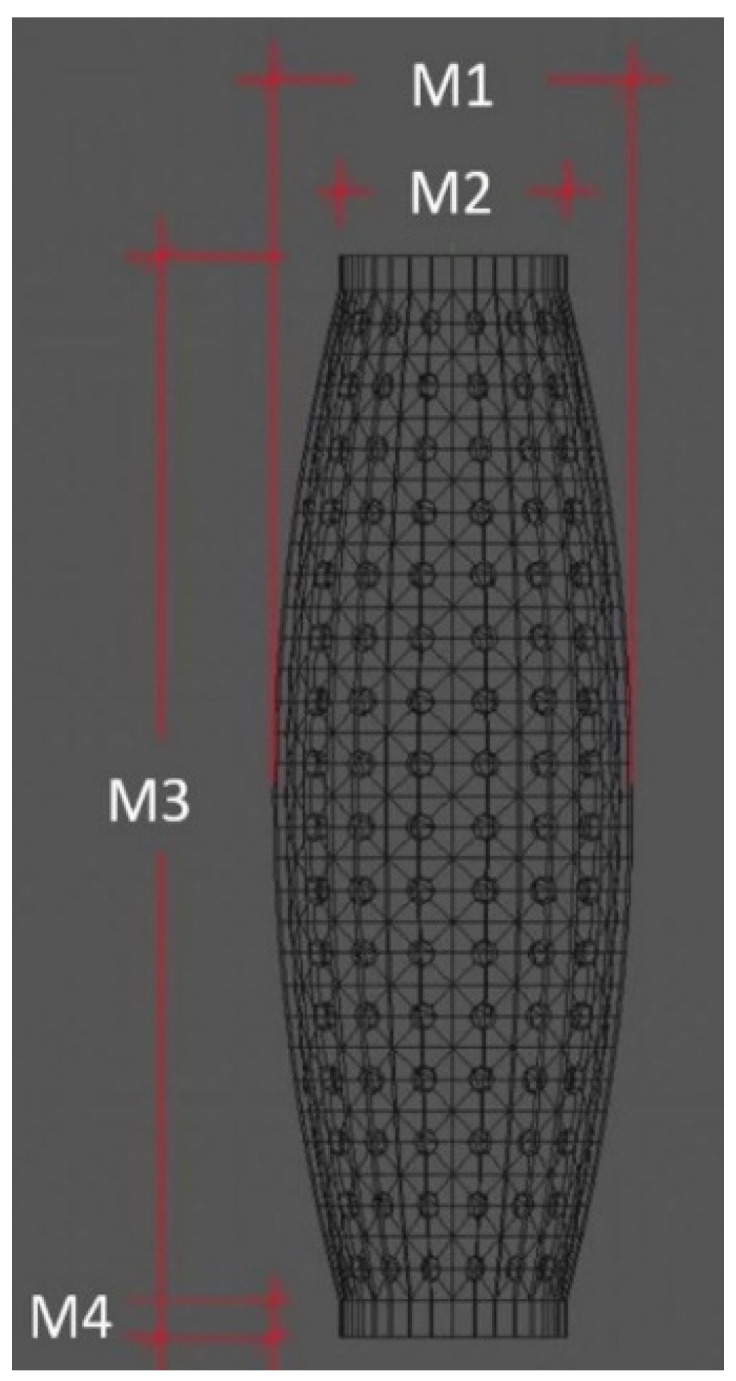
Test Sleeve design.

**Figure 3 sensors-23-00825-f003:**
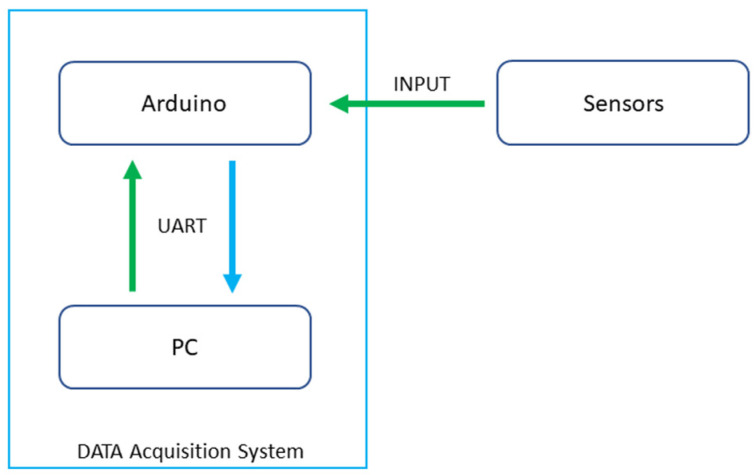
Data acquisition system hardware diagram.

**Figure 4 sensors-23-00825-f004:**
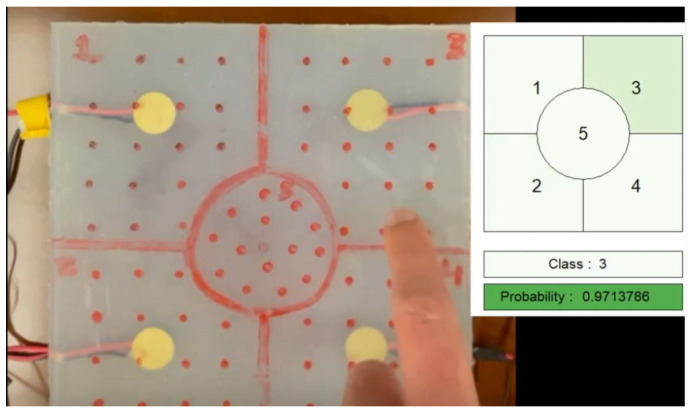
Real-time classification of the prototype.

**Figure 5 sensors-23-00825-f005:**
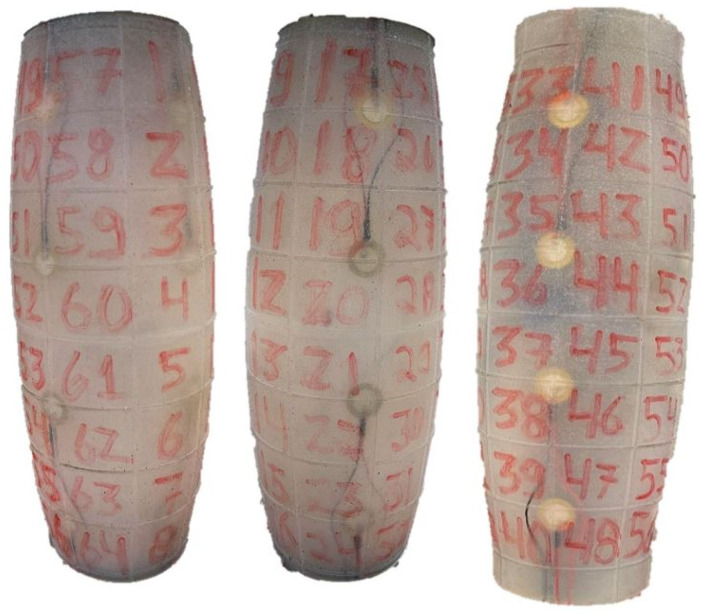
Test Sleeve with piezoelectric sensors (different angles of view).

**Figure 6 sensors-23-00825-f006:**
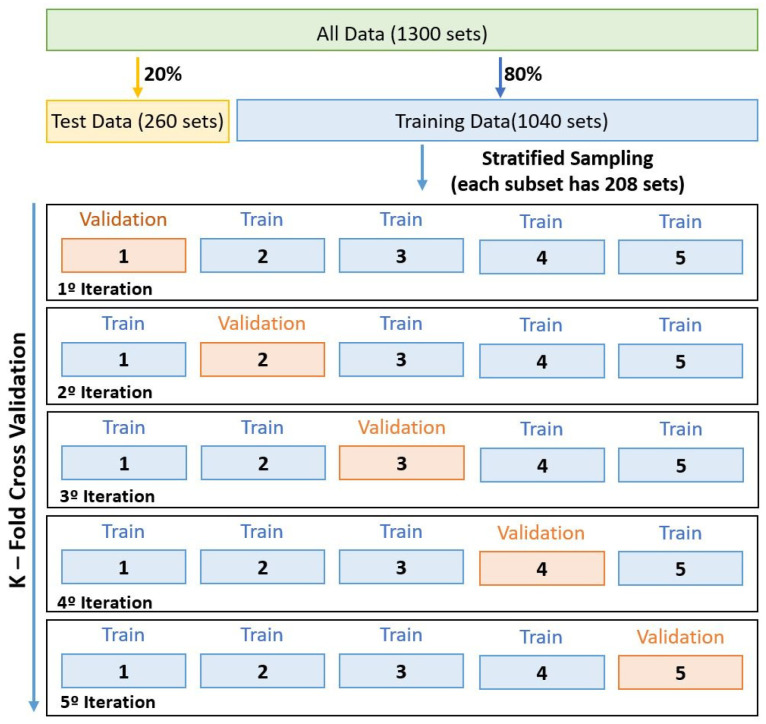
Accuracy of the training and validation data.

**Figure 7 sensors-23-00825-f007:**
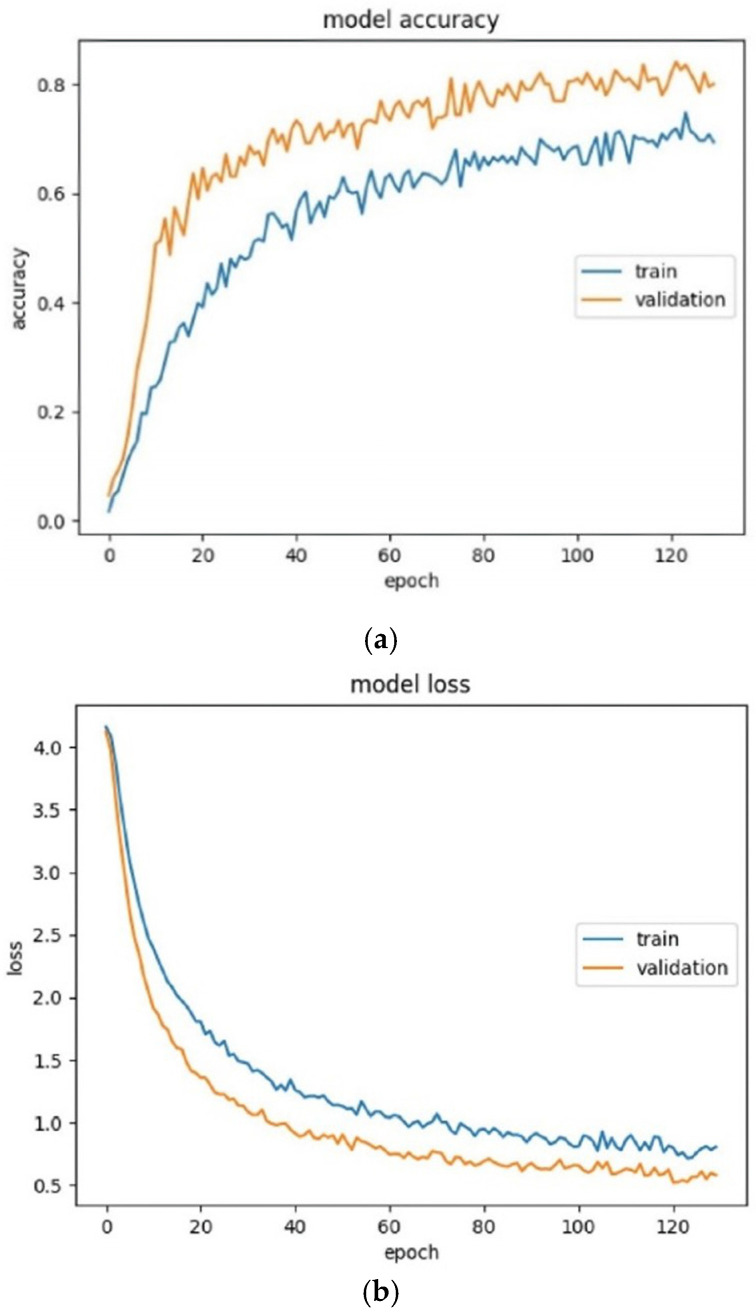
(**a**) Accuracy of the training and validation data; (**b**) Model loss of the training and validation data.

**Figure 8 sensors-23-00825-f008:**
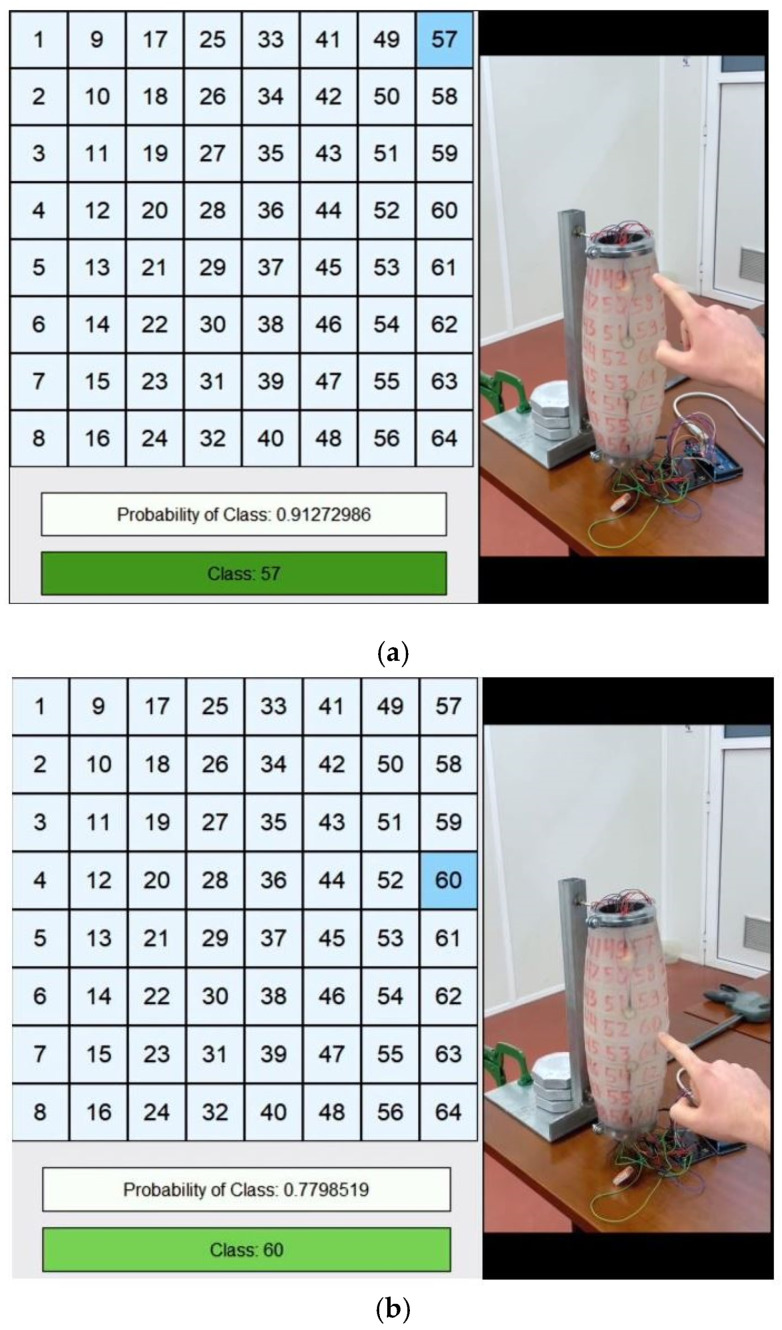
(**a**) Class 57 prediction; (**b**) Class 60 prediction.

**Table 1 sensors-23-00825-t001:** Test Sleeve Dimensions.

Label	Nominal Value (mm)
M1	120
M2	90
M3	340
M4	20

**Table 2 sensors-23-00825-t002:** Detail for each layer of the network.

Type	Output Size
Dense Layer 1	128
Dense Layer 2	64
Dense Layer 3	64
Dense Layer 4	65

**Table 3 sensors-23-00825-t003:** Definition of hyperparameters for the SDN.

Hyperparameters	Output Size
Learning Rate	0.002
Epochs	130
Batch Size	32

## Data Availability

Not applicable.

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
