# Peer review of "Large-Scale Tactile Detection System Based on Supervised Learning for Service Robots Human Interaction"

_sensors, 2023, doi:10.3390/s23020825_

Round 1
Reviewer 1 Report
I have the following specific observations:
1. The use of Computer Vision algorithms to map the surface of the forearm sleeve is not evident. The graphical interface did not require Computer Vision algorithms but only the lighting of predefined regions on a surface according to the classes identified in the ANN. In this interface, the authors forgot to put a square that would identify the non-touch event.
2. I do not understand the role of Figure 1 in the manuscript because it is not used anywhere.
3. Authors should describe how the piezoelectric sensors are interconnected through the microcontroller.
4. More information should be given about the array of sensors in the sleeve -for example the distance between the sensors and the geometry of the array.
5. Some typical sensor values should be given. Were these values normalized before supplying them to the ANN?
6. The architecture of the artificial neural network is unclear. It is identified as ANN. I assume they mean a fully connected Feed Forward neural network.
7. The authors mention, “Since the sensor information is continuous sequential data, the ANNs were the most appropriate structure to recognize the contact zone” I do not understand this statement.
8. The dimensions of the Test Sleeve and the coordinates of the piezoelectric sensors in the sleeve are essential features of the touch sensor. Therefore, they must be integrated into the data supplied to the learning system (ANN).
9. Does the input vector to the ANN contain the information of the 16 piezoelectric sensors? What is the dimension of this vector?
10. What framework was used to build the ANN architecture?
11. The number of data is too small. The time to obtain more data is short compared to writing the manuscript and revising it. A large number of data for the present work, however, is essential.
12. The speed of touching the sleeve should also play a role in the learning system.
13. The quality of Figure 6 should be improved.
14. In Table 3, it is mentioned that the number of epochs was 70, but in Figure 6, the number of epochs is more than 120.
15. From Figure 6, it can be seen that the learning model shows underfitting, although the number of samples is small. How do the authors explain this fact? This figure also shows that the number of epochs can be increased to have better results. Why was the training done in such a small number of epochs? Although the training time was short.
Author Response
Point 1. The use of Computer Vision algorithms to map the surface of the forearm sleeve is not evident. The graphical interface did not require Computer Vision algorithms but only the lighting of predefined regions on a surface according to the classes identified in the ANN. In this interface, the authors forgot to put a square that would identify the non-touch event.
Response 1: The goal of using the computer vision algorithms to map the surface was to replicate the work on the referenced article and use it as a training sleeve for the sleeve intended to be implemented on CHARMIE. It is correct, the graphical interface did not require computer vision algorithms. However, we decided to showcase this anyway, so it is more straightforward for the reader to understand how the detection on the sleeve works. When a non-touch event occurs, none of the 64 squares is highlighted which identifies this event.
Point 2. I do not understand the role of Figure 1 in the manuscript because it is not used anywhere.
Response 2: We added a clearer explanation of figure 1 and its purpose. Line 91-94. "The interface demonstrated in Fig. 1 shows the image captured by the two cameras on both sides of the sleeve, top and bottom and where the algorithm detects a touch event. In addition, on the bottom right it is shown the outside perspective of where the sleeve is being touched. "
Point 3. The authors should describe how the piezoelectric sensors are interconnected through the microcontroller.
Response 3: The piezoelectric sensors are connected to the ADC inputs of the microcontroller. In addition, in parallel with each sensor, there is a 1MΩ resistor in order to limit the current and voltage produced by the piezo sensor to preserve the analog port.
Point 4. More information should be given about the array of sensors in the sleeve -for example, the distance between the sensors and the geometry of the array.
Response 4: The sleeve presents a non-linear cylindrical structure and there is a variation of the perimeter along the sleeve. This particularity caused a variation in the distance of the sensors in the horizontal part of the sensors array, where the ends of the sleeve the sensors are horizontally distant around 5.3cm. In the central area, where the sleeve has the greatest perimeter, the sensors are 8.3cm apart. Vertically, the sensors have a similar distance in all areas of the array, with an average interval of 8cm.
Point 5. Some typical sensor values should be given. Were these values normalized before supplying them to the ANN?
Response 5: The data are normalized to a value between 0 and 1 at the start of the program’s execution, in order to facilitate the model training process. In this conversion, the highest numerical value is obtained from the readings of the 16 sensors, and the remaining values are divided by this value
Point 6. The architecture of the artificial neural network is unclear. It is identified as ANN. I assume they mean a fully connected Feed Forward neural network.
Response 6: On 2.2 Section C there is a description of how the neural network architecture. However, we added for clarity that it is indeed a fully connected feed-forward neural network. Line 167-168
Point 7. The authors mention, “Since the sensor information is continuous sequential data, the ANNs were the most appropriate structure to recognize the contact zone” I do not understand this statement.
Response 7: Since the neural network takes sequential data as input which are the numerical values of the 16 sensors, the most suitable neural network are Recurrent Neural Networks (RNNs). This is because it remembers its inputs due to having an internal memory which makes it appropriate for machine learning problems that involve sequential data.
Point 8. The dimensions of the Test Sleeve and the coordinates of the piezoelectric sensors in the sleeve are essential features of the touch sensor. Therefore, they must be integrated into the data supplied to the learning system (ANN).
Response 8: The goal of the learning system used, is to be independent of the location of the piezoelectric sensors. This way, should a new sleeve be created with a different configuration of sensors, there is no need to calculate or reconfigure any other data. We added this statement to the paper, so it is perceivable that this was our goal.
Point 9. Does the input vector to the ANN contain the information of the 16 piezoelectric sensors? What is the dimension of this vector?
Response 9: The vector has one dimension, but 16 elements that represent each sensor
Point 10. What framework was used to build the ANN architecture?
Response 10: Keras
Point 11. The number of data is too small. The time to obtain more data is short compared to writing the manuscript and revising it. A large amount of data for the present work, however, is essential.
Response 11: As you stated, this is not feasible in the schedule that this special issue presents. However, we will try to still add some tests with more data if possible.
Point 12. The speed of touching the sleeve should also play a role in the learning system.
Response 12: Yes, it should. However, to focus on the machine learning aspect of the project it is assumed (and was tested this way) that all touches on the sleeve had the same speed. The next iteration of this project, is intended to detect the speed and intensity of touching and several other aspects.
Point 13. The quality of Figure 6 should be improved.
Response 13: The quality of both graphs in figure 6 has been enhanced.
Point 14. In Table 3, it is mentioned that the number of epochs was 70, but in Figure 6, the number of epochs is more than 120.
Response 14: It was a mistake!
|
Learning rate |
Epochs |
Batch Size |
|
0,002 |
130 |
32 |
Point 15. From Figure 6, it can be seen that the learning model shows underfitting, although the number of samples is small. How do the authors explain this fact? This figure also shows that the number of epochs can be increased to have better results. Why was the training done in such a small number of epochs? Although the training time was short.
Response 15: The dataset presents 1300 readings from the 16 sensors. For each reading, except for class 0, a force had to be applied manually which is not very accurate. In an ideal case, it would be applied with the same force and location. However, there was no tool available that would allow this test to be done. Therefore, throughout the tests, it was concluded that adding more readings to the set would decrease the model's effectiveness. As for the number of epochs, this hyperparameter was selected based on the results obtained in several tests. As the number of epochs increased, the accuracy plot remained very similar to the one in the figure, However, the performance regressed. Although this is not a significant difference, considering the training time plus the performance, these values were preferred.
Reviewer 2 Report
Introduction
L 32: It is mentioned the robot CHARMIE; but is not mentioned in what contributes to geriatrics.
At the end of the introduction must be mentioned what is the function of the Large-Scale Tactil Detection System.
Materials and Methods
All the images could improve in resolution.
In order to understand the work, it is recommended to add a diagram of the system operation.
L 87 - 93: It must be mentioned the characteristics of the camera used for the vision system.
L 106: Mention the material used for the 3d printing, also whichthe 3d printer used and the software used for the model.
L 139: Mention which microcontroller was used.
L 140: Mention what kind of piezoelectric was used or how it was made.
Results
L 223: Is mentioned that the performance is 80%; justify why this is good.
Conclusion:
What are the main outcome base on the results.
Author Response
Introduction:
L 32: It is mentioned the robot CHARMIE; but is not mentioned in what contributes to geriatrics.
Response 1: A more extensive explanation of how CHARMIE aids in geriatric care has been added (Lines 33-38)
At the end of the introduction must be mentioned what is the function of the Large-Scale Tactile Detection System.
Response 2: The goals of using the Large-Scale Tactile Detection System on CHARMIE have been clarified (Line 43-50)
Materials and Methods:
All the images could improve in resolution.
Response 3: We have enhanced some images that we felt could be enhanced (mainly Fig 6)
In order to understand the work, it is recommended to add a diagram of the system operation.
Response 4: A new image was created
L 87 - 93: It must be mentioned the characteristics of the camera used for the vision system.
Response 5: OV2710 2MP USB CAMERA LOW-LIGHT SENSITIVITY (This information has been added to the main text)
L 106: Mention the material used for the 3d printing, also which the 3d printer used and the software used for the model.
Response 6: The sleeve was not 3D printed. It is made of silicone and the mould was made of PLA. This information has been clarified in Line 115-117
L 139: Mention which microcontroller was used.
Response 7: Arduino Mega
L 140: Mention what kind of piezoelectric was used or how it was made.
Response 8: it is a standard piezo-electric sensor of 2 cm, bought online on https://www.botnroll.com/pt/outros/391-sensor-piezo-electrico.html
Results:
L 223: Is mentioned that the performance is 80%; justify why this is good.
Response 9: This performance is considered good based on all prototypes and tests carried out. This is because three models with several methods were developed in order to make this structure feasible.
Conclusion:
What are the main outcome based on the results. (conclusion)
Response 10: This implementation resulted in a predictive model with an accuracy of 80%, which in real time showed a medium performance, representing by several times the neighbouring sections. Despite not being a desirable result, this model has a huge potential. This is because only 16 sensors were used to cover the entire surface of the sleeve, the extraction of the sensors values to constitute the dataset was not appropriate and only 20 readings per class were used. Even though there are all these limitations, it was possible to obtain a model with reasonable results.
Reviewer 3 Report
Large-Scale Tactile Detection System based on Supervised Learning for Service Robots Human Interaction
Lines 58 to 60:
“This structure consists of a small size printed circuit board shaped like a hexagon, equipped with multiple discrete proximity, acceleration, and temperature sensors”
Why such sizes and shapes? This requires bit explanations.
Lines 127, 128:
2.2. Setup for the Artificial Neural Network (ANN)
128 Unlike the work presented in [11],
This is not right way to write, please elaborate the context for readability of user.
The work is coupled with the CHARMIE robot, a collaborative robot for domestic and medical environments. This system allows the contact detection of the entire forearm surface enabling interaction between a Human Being and a robot.
Work is promising and will be helpful for community.
Abstract: needs revisions to show the required work done and results obtained in the work.
Then towards, related work, less works are discussed, so reader doesn’t get a complete context of the work till now done.
Revise the conclusion, don’t repeat the sentences and please put the closing statements here. About whole of work done and achieved.
Author Response
Lines 58 to 60:
“This structure consists of a small size printed circuit board shaped like a hexagon, equipped with multiple discrete proximity, acceleration, and temperature sensors”
Why such sizes and shapes? This requires bit explanations.
Response 1: The amount and type of sensors are meant to simulate all the sensations produced by human skin. As for the size and shape, they were not very clear on that. But I believe it is to be possible to group a greater number of sensors as well as to have greater effectiveness.
Lines 127, 128:
2.2. Setup for the Artificial Neural Network (ANN)
128 Unlike the work presented in [11],
This is not right way to write, please elaborate the context for readability of user.
Response 2: This has been rewritten so the reader can understand more easily what was meant to be said
The work is coupled with the CHARMIE robot, a collaborative robot for domestic and medical environments. This system allows the contact detection of the entire forearm surface enabling interaction between a Human Being and a robot.
Work is promising and will be helpful for community.
Response 3: Thanks.
Abstract: needs revisions to show the required work done and results obtained in the work.
Response 4: Done.
Then towards, related work, less works are discussed, so reader doesn’t get a complete context of the work till now done.
Response 5: Done.
Revise the conclusion, don’t repeat the sentences and please put the closing statements here. About whole of work done and achieved.
Response 5: Done.
Round 2
Reviewer 1 Report
The revised version has incorporated the answers to the questions about the content of the original manuscript.